# The Link between Autosomal Dominant Polycystic Kidney Disease and Chromosomal Instability: Exploring the Relationship

**DOI:** 10.3390/ijms25052936

**Published:** 2024-03-02

**Authors:** Phang-Lang Chen, Chi-Fen Chen, Hugo Y.-H. Lin, Daniel J. Riley, Yumay Chen

**Affiliations:** 1Department of Biological Chemistry, University of California, Irvine, CA 92697, USA; plchen@uci.edu (P.-L.C.); chifenc@uci.edu (C.-F.C.); 2Division of Nephrology, Department of Internal Medicine, Kaohsiung Medical University Hospital, Kaohsiung Medical University, Kaohsiung 807, Taiwan; hugoyl@gap.kmu.edu.tw; 3Internal Medicine, Kaohsiung Municipal Ta-Tung Hospital, Kaohsiung Medical University, Kaohsiung 807, Taiwan; 4Department of Medicine, College of Medicine, Kaohsiung Medical University, Kaohsiung 807, Taiwan; 5Department of Medicine, Division of Nephrology, University of Texas Health, San Antonio, TX 78245, USA; laufendj@yahoo.com; 6Department of Medicine, Division of Endocrinology, University of California, Irvine, CA 92697, USA

**Keywords:** ADPKD, DNA damage repair, checkpoint, PKD1, PKD2, CHK1, CHK2, ATM

## Abstract

In autosomal dominant polycystic kidney disease (ADPKD) with germline mutations in a *PKD1* or *PKD2* gene, innumerable cysts develop from tubules, and renal function deteriorates. Second-hit somatic mutations and renal tubular epithelial (RTE) cell death are crucial features of cyst initiation and disease progression. Here, we use established RTE lines and primary ADPKD cells with disease-associated *PKD1* mutations to investigate genomic instability and DNA damage responses. We found that ADPKD cells suffer severe chromosome breakage, aneuploidy, heightened susceptibility to DNA damage, and delayed checkpoint activation. Immunohistochemical analyses of human kidneys corroborated observations in cultured cells. DNA damage sensors (ATM/ATR) were activated but did not localize at nuclear sites of damaged DNA and did not properly activate downstream transducers (CHK1/CHK2). ADPKD cells also had the ability to transform, as they achieved high saturation density and formed colonies in soft agar. Our studies indicate that defective DNA damage repair pathways and the somatic mutagenesis they cause contribute fundamentally to the pathogenesis of ADPKD. Acquired mutations may alternatively confer proliferative advantages to the clonally expanded cell populations or lead to apoptosis. Further understanding of the molecular details of aberrant DNA damage responses in ADPKD is ongoing and holds promise for targeted therapies.

## 1. Introduction

Autosomal dominant polycystic kidney disease (ADPKD) is the most common genetic disorder that leads to chronic and end-stage kidney disease. Innumerable cysts form progressively in all parts of the nephron and at different rates in different patients, even those within the same pedigree [1,2]. Initiation of cyst formation in ADPKD is attributed to functional inactivation of polycystin-1 or -2 due to a *PKD1* or *PKD2* mutation. The “two-hit” model first proposed by Reeders in 1992 suggests that each cyst arises from the inactivation of both copies of a *PKD* gene. This involves a germline mutation as the primary event and a subsequent acquired somatic mutation within the epithelial cells lining the cyst [3]. Microdissection studies of individual cysts from PKD patients have supported this model by revealing additional somatic mutations in the *PKD1* or *PKD2* genes [4]. Significantly, these secondary mutations, within the clonal epithelial cells that line individual cysts, differ from the primary germline mutation and are unique in individual cysts from the same patient. These findings indicate that increased mutation rate in tubular cells is a prominent feature of ADPKD. Notably, renal tubular cells exhibit an accelerated rate of spontaneous mutations compared to other somatic cells [5,6]. Renal tubular cells of individuals carrying *PKD1* or *PKD2* mutations are likely more susceptible to DNA damage and aberrant apoptosis, which in turn facilitates spontaneous, cyst-initiating “second-hit” mutations in genes that include *PKD1* and *PKD2*. However, a “third hit” appears necessary for cyst formation. This “third hit” is primarily environmental rather than strictly genetic. Cyst formation is observed during tubulogenesis or following specific injuries that necessitate cellular mitosis (for a comprehensive review, refer to [7,8]). These observations suggest that impairment in DNA damage response (DDR) pathways and aberrant mitosis play crucial roles in ADPKD development. Further evidence supporting the involvement of aberrant DDR in PKD formation arises from studies demonstrating that *Pkd1*^+/−^ mice experience more PKD after ischemia/reperfusion injury, a controlled form of oxidative injury in vivo. These factors strongly indicate impaired injury-repair mechanisms [9,10].

In previous investigations utilizing the comet assay, a technique also known as single-cell gel electrophoresis, peripheral blood lymphocytes from ADPKD patients, when compared to lymphocytes from normal individuals, exhibited significantly higher mean comet tail lengths, indicating abnormally fragmented genomic DNA. This effect was observed before and after exposure to low-dose irradiation (0.5 Gy) [11]. The underlying mechanism behind this abnormal DNA damage-response phenotype in ADPKD, however, has not been explored or explained in detail. In this study, we present compelling evidence demonstrating that cells genetically predisposed to ADPKD owing to *PKD* gene mutations are uniquely defective in repairing DNA. They exhibit severe chromosome breakage, aneuploidy, impairments in the DNA damage response (DDR) pathway, delayed checkpoint responses, compromised DNA repair processes, persistence of DNA damage breaks, and, ultimately, a distinctive transforming phenotype.

## 2. Results

### 2.1. Severe Chromosome Breakage in ADPKD Cells

Thorough examination of metaphase chromosomes obtained from immortalized epithelial cells from various kidney regions and skin fibroblasts of individuals with clinically evident ADPKD revealed the presence of multiple microchromosomes (Figure 1A). In contrast, control cells consisting of immortalized normal diploid renal tubular epithelial cells exhibited minimal chromosomal abnormalities of this nature (Figure 1B). The occurrence of multiple microchromosomes typically arises due to the accumulation of over 100 unrepaired DNA breaks before a cell enters the mitotic phase [12,13,14]. Severe breaks in chromosomes prepared from ADPKD cells (Figure 1B) support this reported phenomenon or pathogenic requirement. Detailed examination of individual chromosomes from the human RTEs showed that more than 50% of mitotic nuclei derived from ADPKD cells displayed multiple chromosome breaks; in contrast, only 3% were observed in similarly immortalized diploid renal tubular epithelial HK2 cells (Figure 1C,D). These findings strongly indicate that ADPKD cells are inefficient in repairing DNA breaks.

### 2.2. Genomic Instability in ADPKD Cells

Given the apparent deficiency in DNA break repair in ADPKD cells, we investigated the consequences of unrepaired DNA damage on chromosome morphology. We hypothesized that if cells fail to properly arrest their division cycles, allowing double-strand breaks (DSBs) or other forms of damaged DNA to be transmitted to daughter cells, then subsequent homologous chromatid pairing would be disrupted. This in turn would lead to gross mutations and chromosomal abnormalities. To test this hypothesis, we examined mitotic chromosome spreads to explore the occurrence of chromosome rearrangements in ADPKD cells. Notably, ring chromosomes were observed frequently in chromosomes prepared from ADPKD cells (Figure 2) but were exceptionally rare in normal cells. To confirm the occurrence of chromosome fusions, we further investigated the centromere and telomere regions of numerous chromosomes. We used human CREST autoimmune serum [17], which specifically binds to proteins associated with the centromere region, and telomere repeat binding factor 1 (TRF1) [18], which binds to telomeric DNA, to label the centromeres and telomeres of the chromosomes (Figure 2B). Centromere regions were frequently absent in chromosomes carrying *PKD1* mutations (Figure 2B, upper panels, indicated by red arrows). Moreover, chromosome fusions were common, as manifest by the presence of TRF1 at locations that should correspond to centromeres (Figure 2B, lower panels, indicated by green arrows). Fluorescence in situ hybridization (FISH) analysis using a telomeric DNA probe was performed to confirm the instability of telomere regions. This analysis revealed the absence of telomeric DNA at the ends of chromosomes prepared from ADPKD cells and its detection in inappropriate positions, such as the central regions of chromosomes. Our observations in mitotic figures strongly suggest the occurrence of complex chromosome rearrangements in ADPKD cells (Figure 2C, indicated by arrows).

We conducted further FISH analysis to validate the occurrence of chromosome rearrangements in ADPKD cells. Probes targeting the Smith–Magenis region (17P11.2, located close to the centromere) and the Miller–Diek-r/ILS region (17P13.3 near the telomere) of human chromosome 17 were utilized. These regions typically appear in normal cells as two dots representing the two alleles (Figure 2D). In ADPKD cells, however, we frequently observed abnormalities in the hybridization patterns. These abnormalities included the presence of only one allele (single dot) or more than two alleles (multiple dots) hybridizing to either the centromere (green) or telomere (red) region. These results strongly indicate the occurrence of chromosome rearrangements, such as loss or loss accompanied by amplification, in ADPKD cells.

### 2.3. Defective DNA Damage Response in ADPKD Cells

Severe chromosome breakage and chromosome rearrangement strongly indicate the presence of defective DNA damage responses and/or impaired DNA repair mechanisms in ADPKD cells. To investigate whether ADPKD cells exhibit defective DNA damage responses, we assessed their ability to activate known checkpoint pathways in response to DNA damage induced by UV or gamma irradiation. For this analysis, we utilized an antibody specific to the phosphorylated histone H2A variant X (γ-H2AX), a marker for DNA breaks that allows reliable detection of damaged DNA [19]. Initially, we examined DNA damage events in untreated ADPKD cells. Many untreated ADPKD cells exhibited γ-H2AX foci, whereas such detection was rare in wild-type cells. This finding suggests that DNA damage occurs at higher-than-regular rates in ADPKD cells, indicating a potential impairment in their DNA damage-response mechanisms.

To investigate the response of ADPKD cells to DNA damage, we examined the activation of essential DNA damage-response sensor proteins, ATM and ATR [20], as well as the phosphorylation of cell cycle checkpoint kinases Chk1 and Chk2 [21], using Western blotting analysis. In normal cells, ATM, ATR, CHK1, and CHK2 are activated through phosphorylation at specific residues (Ser1981 for ATM, Ser428 for ATR, Ser345 for CHK1, and Thr68 for CHK2), and their activation in response to DNA damage can be detected using phospho-specific antibodies.

We found that ATM and ATR activations remained intact after DNA damage in ADPKD cells (Figure 3). The more distal checkpoint protein kinases, CHK1 and CHK2, however, failed to be activated in ADPKD cells, in contrast to their normal activation in the HK2 diploid RTEs used as controls. Although the Western blotting analysis detected S-1981 phosphorylation and activation of ATM in ADPKD cells, the accumulation of ATM at nuclear foci, which are sites of damaged DNA, was not detectable in these same cells after treatment with ionizing radiation (IR) (Figure 3C). Therefore, even the upstream sensing event of complete and timely ATM recruitment appears to be aberrant in ADPKD cells in response to the stress of IR-induced DNA damage. 

WT9-7 and WT9-12 are human renal tubular epithelial cell lines carrying *PKD1* mutations [16]. These cell lines were initially established through immortalization using the simian virus 40 (SV40) T antigen, a viral oncoprotein [22]. Since SV40 T antigen has been shown to interfere with the proper activation of DNA damage checkpoints [23], the use of virally transformed cells introduces a theoretical concern regarding the unwanted effects of the oncoprotein, separate from the effects arising from the underlying *PKD1* mutations.

To try to separate the potential influence of the T antigen on the checkpoint defects we observed in WT9-7 and WT9-12 cells from the more relevant effects of the underlying *PKD* gene mutations and polycystin protein abnormalities, we performed similar experiments in primary, untransformed ADPKD cells. We first established a line of untransformed RTE cells called 597A. These cells were cultured from a nephrectomy specimen from an individual with late-stage ADPKD and could be propagated for up to six passages. Targeted sequencing conducted by a commercial lab (Athena Diagnostics, Worcester, MA, USA) determined the unique genetic variant in these cells to be a C11390G transversion in exon 39 of *PKD1*. This transversion results in a missense mutation (Ala3727Pro) in one of the transmembrane regions of polycystin-1 which is not found in the PKD foundation or LOVO global virome data banks. 

Initial observations in 597A cells were similar to those in WT9-7 and WT9-12 cells. Following exposure to ionizing radiation, IRIF of ATM failed to be detected within 1 h, although phosphorylated ATM (S1987) was detected by Western blotting shortly after irradiation (Appendix A). Unlike WT9-7 (Figure 3C) and WT9-12 cells, however, phosphorylated ATM in 597A cells was localized in the cytoplasm (Figure 4A). Similar results were obtained in 1096Sk (Appendix A).

To investigate whether phosphorylated ATM correctly enters the nuclei and marks DNA damage sites over time, we exposed cells to ionizing radiation (IR). We then fixed them at different points in time after IR treatment. In 597A and 1096Sk cells, the appearance of ATM immunoreactive foci (ATM IRIF) was significantly delayed. By 3 h after IR in 597A cells and by 6 h after IR in 1096Sk cells, nearly 100% of the cells did exhibit ATM IRIF (Figure 4A,B and Appendix A). Activation of CHK2 also exhibited a similar temporal delay, coinciding with the formation of ATM IRIF (Figure 4A,B). It is important to note that this delayed DNA damage response is not specific to kidney epithelial cells; ADPKD skin fibroblasts also displayed a similar delay in the formation of ATM IRIF compared to normal, untransformed human skin fibroblasts (Figure 4C and Appendix A).

This delay in the typical DNA damage response may explain the observed failure of proper cell cycle arrest in 597A cells. These cells failed to arrest at the G2/M phase after IR treatment (Figure 4D) and exhibited increased sensitivity to the lethal effect of IR compared to HK2 cells that were treated identically (Figure 4E).

### 2.4. DNA Damage in Autosomal Dominant Polycystic Kidneys

The abnormal localization of checkpoint proteins may explain the severe chromosome breaks observed in cultured ADPKD cells. However, it is essential to determine if this phenotype is an artifact of in vitro culture. To extend our findings beyond cultured cells, we examined DNA checkpoint activation, unrepaired DNA, and resulting chromosome damage in ADPKD using human kidney specimens. We employed immunohistochemistry (IHC) to assess DNA damage in situ in 14 formalin-fixed, paraffin-embedded ADPKD nephrectomy samples.

Initially, we used antibodies that specifically recognize 8OHdG (indicative of oxidative lesions) and anti-γ-H2AX (indicative of DNA breaks) [19,24,25]. All samples exhibited numerous characteristic foci of 8OHdG and γ-H2AX in the nuclei of tubular epithelial cells, including cells lining both histologically normal tubules and those lining cysts (Figure 5A,B and Table 1). These IHC results indicate the presence of severe DNA damage in RTEs in advanced-stage kidneys removed from patients with ADPKD.

Next, we performed IHC on histologic sections from the same tissues using antibodies that reliably detect activated ATM, ATR, CHK1, CHK2 (Figure 5C–F), and other human DNA damage-sensing and repair-pathway proteins. Consistent with the observations in cultured ADPKD cells, the activation of checkpoint proteins appeared to be impaired in the affected cells (Figure 5C–F and Table 1). These in situ staining results validate the findings in cultured cells and provide further support for a broader problem in ADPKD—namely, the failure of the normal DNA damage response leading to excessive unrepaired DNA.

ADPKD kidney sections were subjected to immunohistochemistry staining using antibodies against various DNA damage-response/repair proteins. The staining targets included early-response sensor proteins, activated ATM, ATR, and Nek1, mediators such as MDC1/NFBD1 and 53BP1, repair proteins including Mre11, NBS, and Rad50, and checkpoint proteins, as well as activated CHK1, CHK2, and ATM substrates (pSQ).

### 2.5. DNA Breaks Persist in ADPKD Cells

To investigate whether the abnormalities in DNA damage response and checkpoint control directly contribute to the defective DNA repair observed in ADPKD, we examined the DNA repair ability of relatively normal diploid renal tubular epithelial (RTE) cells (HK2) and ADPKD cells after low-dose ionizing radiation (IR). We utilized γ-H2AX nuclear foci as an indicator of DNA break repair in the cells, scoring such foci at different time points following DNA damage.

One hour after IR, both HK2 and ADPKD cells displayed γ-H2AX immunoreactive foci (IRIF), indicating the presence of DNA breaks in both cell types (Figure 6A,B). However, 24 h after IR, HK2 cells successfully repaired their DNA breaks, resulting in the absence of γ-H2AX nuclear foci. In contrast, WT9-7 and WT9-12 ADPKD cells failed to repair the DNA breaks, with nearly 100% still exhibiting γ-H2AX nuclear foci even 24 h after exposure to a very low dose of IR (1 Gy). Similar observations were made in primary 597A ADPKD cells, which were never immortalized with any viral oncoprotein. Following a slightly higher dose of IR (2.5 Gy), almost 100% of all 597A cells still displayed γ-H2AX foci after 24 h, indicating their failure to repair damaged DNA (Figure 6C). On individual cell nuclei, 24 h after low-dose IR (1 Gy), we performed comet assays, to further confirm our findings. ADPKD cells exhibited significantly longer and denser comet moments, which indicate fragmented DNA. These results provide additional evidence of excessive and unrepaired DNA double-strand breaks (DSBs) (Figure 6D,E and Appendix A).

These results demonstrate that non-lethal DNA damage persists for an unusually long time after IR in ADPKD cells, indicating the cell’s compromised ability to repair DNA breaks. This defective DNA repair is consistent with the abnormalities observed in DNA damage response and checkpoint control pathways in ADPKD, and further highlights the underlying DNA repair deficiency in ADPKD cells.

### 2.6. Transforming Phenotypes of ADPKD Cells

The defective DNA damage response and repair observed in ADPKD cells raise the possibility that they may exhibit other characteristics commonly found in cells with defective DNA repair, like cancer cells. To explore this hypothesis, we investigated additional growth characteristics, such as loss of contact inhibition and increased saturation density, in ADPKD cells and wild-type control cells.

Control HK2 cells displayed a significantly lower saturation density (0.9 × 10^5^/cm^2^) compared to the two ADPKD cell lines, WT9-12 (5.5 × 10^5^/cm^2^) and WT9-7 (2.8 × 10^5^/cm^2^) (Figure 7A). The observed phenotypes in ADPKD cells prompted us to explore other features associated with neoplastic behavior. A reliable indicator of the ability of cells to form tumors in vivo is their capacity for anchorage-independent colony formation in soft agar. Therefore, we employed the soft agar assay to assess the ability of ADPKD cell lines (WT9-12 and WT9-7) to form colonies without attachment to plastic. Normal RTE cell lines (human HK2 and monkey BS-C1) served as negative controls, while human renal cell carcinoma (RCC) cell lines (786-O and A498) served as positive controls. Cells were seeded into soft agar, and their ability to form colonies was evaluated. Like RCC cells, ADPKD cells formed colonies in the soft agar, indicating anchorage-independent growth. In contrast, normal cells exhibited poor growth in the same assay (Figure 7B,C). These findings suggest that ADPKD cells can acquire a transformed phenotype. 

Although renal cell carcinoma with metastatic potential is probably not more common in ADPKD patients than in the general population with chronic kidney disease [28,29], several reports have documented a high incidence of renal adenomas and carcinomas in nephrectomy and autopsy specimens from ADPKD patients [30,31,32,33]. Our thorough examination of the 14 ADPKD nephrectomy specimens showed multiple adenomas, predominantly exhibiting papillary characteristics, in most of these late-stage polycystic kidneys (Figure 7D). This observation and similar findings reported by other researchers further support our claim that ADPKD cells are prone to transformation.

These findings suggest that ADPKD cells exhibit features associated with transformed or neoplastic behavior, such as increased saturation density, anchorage-independent growth, and renal adenomas. These characteristics may arise due to the defective DNA repair mechanisms observed in ADPKD cells, indicating potential implications for developing renal tumors in ADPKD patients.

## 3. Discussion

This study presents compelling evidence that severe DNA breakage, impaired DNA damage response, and impaired DNA damage-repair mechanisms are important pathogenic mechanisms in the generation or progression of autosomal dominant polycystic kidney disease. While the early DNA damage-response protein kinases, ATM and ATR, show intact activation in ADPKD cells, as indicated by the detection of specific phosphorylation events upon activation, the downstream checkpoint protein kinases, such as CHK1 and CHK2, fail to activate as expected. Furthermore, we observed a delay in the localization of activated ATM at DNA damage sites in ADPKD cells, which correlated with the delayed activation of CHK2. These abnormalities in propagating DNA damage sensing have significant implications: ADPKD cells exhibit compromised DNA repair capabilities, as evidenced by severely damaged DNA that persists over time. In contrast, normal kidney epithelial cells exhibit minimal or undetectable damaged DNA.

Interestingly, the presence of damaged DNA is not solely attributed to external factors such as exposure to ionizing radiation (IR) or ultraviolet (UV) light. Instead, our findings indicate that damaged DNA is a spontaneous phenomenon in ADPKD renal tubular cells. It occurs and recurs during cell proliferation. These findings shed light on the underlying mechanisms contributing to the accumulation of DNA damage in ADPKD cells and suggest a potential link between defective DNA repair and the pathogenesis of ADPKD. Further investigation is required to elucidate the precise molecular mechanisms underlying these abnormalities and their implications for the development and progression of ADPKD.

The discovery of excessive DNA damage in ADPKD cells, even without specific external damage such as radiation, aligns with the findings of a previous study investigating DNA stability in lymphocytes from ADPKD patients and healthy individuals [11]. Compared to age-matched healthy controls, lymphocytes from ADPKD patients exhibited higher levels of spontaneous DNA damage. These results, combined with our findings in renal and skin cells, suggest that the DNA instability observed in ADPKD is not limited to a specific tissue type. Instead, the aberrant DNA damage responses and defective DNA repair mechanisms in ADPKD likely arise from the impaired function of polycystin-1/2, which we propose to be directly or indirectly involved in the DNA damage-repair pathway. This suggests a broader impact of polycystin dysfunction on genomic stability and highlights the need for further investigations to elucidate the specific mechanisms underlying these observations.

The kidneys are particularly affected in patients with germline mutations in *PKD1* or *PKD2*, which can be attributed to several factors. One possible reason is that the kidneys are organs responsible for filtering toxins from the body. Various common and toxic insults, such as exposure to intravenous contrast, ischemia/reperfusion injury, and a high-fat/high-salt Western-type diet, can lead to increased production of reactive oxygen species (ROS) or oxygen free radicals in the kidneys. These ROS can cause damage to renal epithelial cells through multiple mechanisms, leading to necrosis or apoptosis. If DNA damage is not repaired properly or in a timely manner, then some cells may not survive when they divide. Other cells with less lethal DNA damage might acquire advantages in proliferation, such that they clonally expand. Several studies have shown that cells lining individual cysts in kidneys from humans with ADPKD are indeed clonal, and that the second ”hit” beyond the original *PKD1* or *PKD2* gene mutation is different within the same kidney [4,6]. Severe injury can result in mitochondrial dysfunction, ATP depletion, cytoskeleton disruption, cell polarity loss, impaired directional solute transport, lipid peroxidation, DNA damage, and other detrimental effects.

Animal models of ADPKD have provided insights into the development of cysts, which require cellular proliferation and inactivation of genes involved in ADPKD during embryonic or neonatal kidney development or in response to injury and repair processes [34]. However, if the proteins involved in ADPKD are conditionally inactivated after complete kidney maturation, and if the adult kidneys remain uninjured to the extent that renal cells are not forced to undergo division, the development of cysts is rare. In animal models of PKD, ischemia/reperfusion injury, which represents in vivo oxidative injury and includes oxidative DNA damage, has been shown to exacerbate PKD [35]. This observation strongly supports the idea that oxidative DNA damage contributes to the progression of PKD. Overall, the susceptibility of the kidneys to phenotypic manifestations in PKD may be attributed to their role in toxin filtration, the generation of ROS in response to toxic injuries, and the complex interplay between cellular proliferation, gene inactivation, and injury/repair processes.

Although genomic instability is a hallmark of cancer, the relationship between PKD and cancer has yielded conflicting findings in various studies. Clinical studies conducted on patients after renal transplantation have shown that the incidence of renal cell carcinoma (RCC) is not significantly increased in ADPKD patients compared to non-ADPKD patients [36,37,38]. One recent study by the US Transplant Registry even reported a decreased incidence of RCC in kidney transplant recipients with ADPKD compared to recipients whose end-stage kidney disease was caused by other types of tubule-interstitial and glomerular disease [39]. It is important to note that these studies primarily focused on the post-transplantation period, and that the evaluation of cancer incidence before renal transplantation was not performed. The conclusions drawn from these studies could be limited by the inability to control for the effects of immunosuppression following transplantation. Immune suppression can have significant and unpredictable effects on the initiation and progression for RCC, in both its clear-cell and papillary types.

One large-scale clinical study conducted in Taiwan involving 4346 ADPKD patients and 4346 non-ADPKD patients focused on cancer without the influence of transplant immunosuppression. It revealed a higher incidence of cancer in ADPKD patients, particularly liver, colon, lung, and kidney cancer [40]. Notably, these organs are involved in removing toxic waste from the body, which may result in increased exposure to genomic insults. When considering the defective DNA damage response and repair mechanisms observed in our present study, it is plausible to suggest that the higher incidence of cancer found in ADPKD patients compared to non-ADPKD patients could be attributed to compromised DNA repair mechanisms when patients have underlying germline *PKD1* or *PKD2* mutations. The conflicting studies regarding cancer incidence in patients with ADPKD, when taken along with our findings, highlight the complex relationship between PKD, genomic instability, and cancer incidence. Further research to elucidate the underlying mechanisms and to understand better the interplay between PKD, DNA damage response, and cancer development is needed.

In the DNA-damage-checkpoint control system, four main components are involved: sensors, mediators, transducers, and effectors [41]. Sensors are responsible for recognizing DNA damage and initiating subsequent events. Critical sensors include ATM, ATR, Nek1 protein kinases, and the RFC/PCNA-related Rad17-RFC/9-1-1 complex. Mediators interact with sensors and transducers to ensure specific signal transduction. Some examples of mediators are 53BP1, NFBD1/MDC1, BRCA1, and claspin. Transducers, such as CHK1 and CHK2, are phosphorylated by ATM or ATR, enabling them to activate downstream effectors involved in cell cycle arrest (p53 and cdc25) and DNA repair (such as Mre11/NBS/Rad50). These effectors directly activate checkpoint control and DNA damage-repair mechanisms. In our analysis of human clinical specimens, we examined the expression of sensors (ATM, ATR, Nek1), mediators (53BP1, MDC1), transducers (CHK1, CHK2), and effectors (Mre11). We found excessive DNA damage in all the samples, as evidenced by the formation of γH2AX nuclear foci. While the tested sensors (ATM/ATR) were activated, they failed to localize to the DNA damage sites (at nuclear foci). Instead, almost all clinical samples showed activated ATM and ATR localized in the cytosol, suggesting a possible defect in the transportation of ATM and ATR to the nuclei in ADPKD. This finding implies that DNA damage signaling initiates in the cytosol and then transduces to the nucleus. 

In contrast to ATM/ATR, the sensor protein Nek1 and its interacting protein 53BP1 were translocated into nuclear foci in most clinical samples. Notably, despite 53BP1 activation, CHK1 and CHK2 transducers were not activated in most samples. Additionally, the downstream effector Mre11 was absent in nuclear foci in most samples, with nearly half showing undetectable levels of Mre11. Genes involved in DNA damage response (DDR) pathways have been shown to be upregulated in ADPKD [42]. However, the localization of these proteins was not at the DNA damage sites, providing a plausible explanation for the failure of DNA repair observed in ADPKD cells. This failure may contribute to the accumulation of DNA breaks in ADPKD cells.

DDR pathways have emerged as a promising therapeutic target in autosomal dominant polycystic kidney disease (ADPKD) [43]. Notably, applying pharmaceutical ATM inhibitors for treating ADPKD cells in vitro has demonstrated encouraging results [44]. Insights from an animal model featuring altered loci for both PKD1 and ATM (Pkd1^RC/RC^/Atm^+/−^ or Pkd1^RC/RC^/Atm^−/−^) revealed no discernible alteration in the progression of cystic kidney disease, even at the age of 6 months [44]. This finding substantiates our observation that the dysfunctional ATM signaling resulting from altered PC1 function challenges ATM’s viability as a potential therapeutic target for ADPKD.

Our investigation sheds light on a potential mechanism that may explain the heightened occurrence of spontaneous mutations particularly in renal tubular cells compared with most other somatic cells [5,6]. Notably, despite the heterozygosity for *PKD1* in WT9-7 and the homozygosity in WT9-12, both cell lines exhibited identical defects in DDR pathways, indicating haploid insufficiency of polycystin-1. Further details to explain how mutations in *PKD1* contribute to defective DDR pathways necessitate further exploration. Studies in this direction are ongoing because understanding these mechanisms is crucial for advancing our knowledge of ADPKD pathogenesis—and for targeted therapeutic interventions beyond the antihypertensive and vasopressin-2 receptor antagonists, they are the best current pharmaceutical agents to slow the progressive decline of renal function in patients with ADPKD.

In summary, we have uncovered evidence suggesting that individuals with autosomal dominant polycystic kidney disease (ADPKD) heterozygotes have haploinsufficiency, and with it an impaired capacity to repair DNA damage. This compromised repair state essentially constitutes the second hit, making nullification of the remaining wild-type more likely than usual—and different in each unique cyst within a polycystic kidney. Interestingly, our findings suggest that both the second and third hits required for full manifestation of renal cysts and progressive renal dysfunction may stem from similar threats. Kidney tubular cells, because of their high metabolic rates and constant filtration functions, face unique challenges. This “third hit” may manifest in various forms beyond just the need for cell division that has been shown in mouse models with inactivated *Pkd1* genes [8]. For instance, compromised repair mechanisms, as observed in conditions like diabetes mellitus, disrupt cellular metabolic homeostasis, leading to dysregulated DNA repair pathways and subsequent genomic instability [27]. Other unique environmental factors, such as the high NaCl-induced exclusion of Mre11 from the nucleus, pose additional hurdles for tubular epithelial cells [45]. Furthermore, hazardous substances like BPA, which are filtered by the kidneys and concentrated in tubular filtrate, pose heightened risks to individuals with compromised repair capacity [46]. Therefore, our observation of defective DNA damage response (DDR) in ADPKD not only sheds light on disease development but also identifies pertinent harmful factors. This insight provides potential avenues for improved care aimed at delaying the progression of ADPKD.

## 4. Materials and Methods

Cell culture. Human proximal renal tubular epithelial cells (HK2, WT9-7, and WT9-12), as well as skin fibroblasts (CCD-986Sk and CCD-1096Sk), were obtained from the American Type Tissue Collection (Rockville, MD, USA). Both CCD-986Sk and CCD-1096Sk represent untransformed skin fibroblasts originating from a normal (CCD-986Sk) or distinct ADPKD patient (CCD-1096Sk). The mutations identified in CCD-1096Sk include variants at the PKD1 locus (A2427G—T1739R, T12039C—L3943P) and the PKD2 locus (G149C—R28P). These cells were cultured in a 1:1 mixture of Ham’s F-12 and Dulbecco’s modified Eagle medium, supplemented with 25 mM HEPES, 10% fetal bovine serum, and antibiotics. The ADPKD 597A cells were established from cortical fragments of a fresh nephrectomy specimen obtained from a female patient with chronic kidney disease stage 5. The patient’s kidneys were removed before a pre-emptive kidney transplant. The procedure for establishing primary renal tubular epithelial (RTE) cells has been previously described [47]. The primary RTE cells, used in passage 2 to 4, were cultured in the same medium as the HK2 cells. 

Ionizing and ultraviolet radiation. Cells were exposed to γ-irradiation using Cesium 40 as the radiation source at a 116 cGy/min rate. The irradiation process was carried out under controlled conditions. Cells were washed with phosphate-buffered saline (PBS) twice and exposed to UV irradiation to remove any external contaminants. Subsequently, the cells were placed inside a UV crosslinker (Stratagene, La Jolla, CA, USA). The dose of UV irradiation was carefully monitored using a UV meter to ensure accurate measurement and control of the UV dose applied to the cells. This step helped maintain consistency and reproducibility in the experimental procedure.

Antibodies. Rabbit polyclonal anti-Nek1 antibodies, which have been previously described and thoroughly characterized [47], were utilized in this study. To detect p84 protein, the monoclonal antibody 5E10 was purchased from GeneTex (Irvine, CA, USA). To detect phosphorylated ATM at serine 1981 (S1987 in mouse), antibodies were obtained from Rockland Immunochemicals (Gilbertsville, PA, USA). Antibodies against γ-H2AX, phosphorylated L-S/T-Q, phosphorylated Chk1 at serine 345, phosphorylated Chk2 at threonine 68, phosphorylated ATR at serine 428, histone H3, and phosphorylated serine 10 of histone H3 were also obtained from Cell Signaling Technology (Danvers, MA, USA). These antibodies were used to facilitate the detection and analysis of specific proteins and phosphorylation events in the experimental samples.

Immunoblot analysis. To extract cellular proteins, cells suspended in Lysis 250 buffer underwent three freeze/thaw cycles alternating between liquid nitrogen (−196 °C) and a 37 °C water bath. Subsequently, the samples were centrifuged at 14,000 rpm for 2 min at room temperature, following a previously established protocol [48].

Scoring for immunofluorescent nuclear foci at sites of DNA damage. A cell was considered positive for immunofluorescent nuclear foci (IRIF) if it exhibited a minimum of 5 distinct fluorescent dots per nucleus. To determine the percentage of nuclei positive for IRIF, the number of nuclei displaying five or more IRIF was divided by the total number of nuclei stained with DAPI in the corresponding microscopic fields. This method has been previously validated and described extensively in publications [49], providing a reliable and standardized approach for quantifying IRIF in the experimental analysis.

Mitotic checkpoint. RTE cells were exposed to γ-irradiation at a dose of 4 Gy. Subsequently, the cells were fixed and subjected to immunostaining using anti-phospho-H3 antibodies at various time points following irradiation. Images of the cells were captured using a fluorescence microscope, and the numbers of phospho-H3-positive cells were quantified as percentages of the total cell counts.

Chromosome spreads and breaks. Cycling cells were exposed to colchicine (1 μg/mL, from Sigma, St. Louis, MO, USA) for 30 min at 37 °C. Following this treatment, all cells, including those present in the supernatant, were collected by trypsinization. The collected cells were then subjected to swelling in a solution of 75 mM KCl for 15 min at 37 °C. Subsequently, the cells were fixed using a freshly prepared mixture of methanol and acetic acid in a ratio of 3:1. The resulting fixed cells were used to prepare chromosome slides, and these slides were stained with Giemsa for visualization. To analyze chromosome breaks, cells were first incubated with hydroxyurea (0.1 μM, from Sigma) for 24 h before the colchicine treatment. This experimental setup allowed for the examining of chromosome structure and evaluating chromosome breakage in response to the combined treatments.

Immunohistochemistry. Human kidney samples for immunohistochemistry (IHC) analysis were obtained either during organ implantation/revascularization or at the time of removal. The collected specimens were fixed by immersion in 10% neutral buffered formalin overnight. Following fixation, the samples underwent a series of dehydration steps and were embedded in paraffin. Thin sections measuring three μm thickness were prepared from the paraffin-embedded tissue blocks. These sections were stained with Meyer’s hematoxylin and eosin reagents to visualize the cellular morphology.

Sections of kidney tissue measuring four μm were deparaffinized on slides, using Histoclear (National Diagnostics, Atlanta, GA, USA) for immunohistochemical staining and rehydrated through a series of graded ethanol solutions. The sections were then treated with 0.05% saponin (Sigma) in distilled water (ddH_2_O), followed by blocking with 10% normal goat serum for 30 min at room temperature. Primary antibodies were added to the sections and incubated overnight at 4 °C. After thorough washing with phosphate-buffered saline (PBS), biotin-labeled secondary antibodies specific to rabbit IgG were applied, followed by the application of an immuno-peroxidase-based ABC (avidin-biotin complex) development kit (Vector Laboratories, Burlingame, CA, USA). Color development was achieved, and the sections were counterstained with methyl green to visualize the nuclei. Our Institutional Review Board specifically approved using excess or discarded, deidentified human kidney specimens.

Comet Assay. Cultured cells were subjected to the Comet assay’s single-cell gel electrophoresis (SCGE) under alkaline conditions [50,51]. The CometAssay™ kit from Trevigen, Inc. (Gaithersburg, MD, USA) was utilized for this purpose. Nuclei within the cells were stained with SYBR Green and analyzed using a Zeiss fluorescence microscope with a digital camera. Multiple random fields were photographed to capture the nuclei.

For each experimental condition, more than 90 individual nuclei were examined. Various parameters such as comet tail length, intensity, percentage of DNA in the tail relative to the head, and the tail area were measured for each nucleus. These measurements were used to calculate tail moments and olives, indicators of DNA damage, using the CometScore™ 1.5 software (available at http://autocomet.com/products_cometscore.php (accessed on 20 August 2020)). The calculated comet tail moments were normalized to the control group, which refers to the mean tail moment of wild-type cells without any exposure to ionizing radiation (IR). The results were presented in histograms as fold increases compared to the control group, indicating the extent of DNA damage.

## 5. Conclusions

Our findings strongly support the notion that the complete and timely activation of the DNA damage signaling pathway relies on the proper functioning of various components, including sensors, mediators, transducers, and downstream effectors responsible for cell cycle arrest and DNA damage repair. Importantly, our results highlight the involvement of PC1/PC2 in the DNA damage response and repair pathway. Understanding the precise roles of PC1/PC2 in DNA damage repair could hold significant therapeutic potential for treating or preventing ADPKD progression. Further investigation in this area is warranted to unravel the molecular mechanisms underlying the contribution of PC1/PC2 to DNA damage repair and to explore potential therapeutic strategies targeting this pathway in ADPKD.

## Figures and Tables

**Figure 1 ijms-25-02936-f001:**
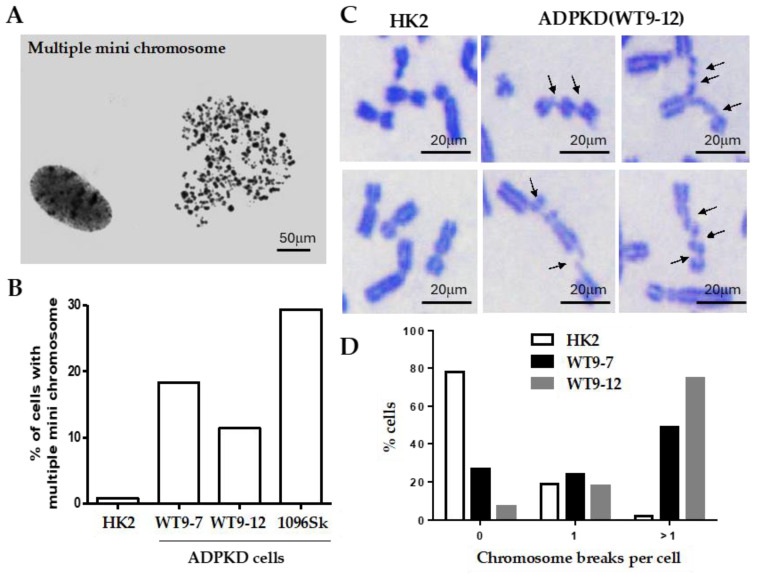
Severe chromosome breakage in ADPKD cells. (**A**,**B**) Diploid HPV16 E6/E7 transformed renal tubular epithelial cells (HK2) [15], SV40-T antigen transformed ADPKD renal tubular epithelial cells (WT9-7, WT9-12) [16], and untransformed ADPKD skin fibroblast cells CCD-1096Sk (ATCC-CRL2129) (1096Sk) were treated with nocodazole (0.4 μg/mL) to enrich the population of mitotic cells. Free chromosomes were released from the lysed cells and then dispersed and stained with Giemsa. (**A**) A representative chromosome spread of multiple microchromosomes from a 1096Sk cell was observed. These indicate the presence of chromosomal abnormalities. (**B**) Percentages of cells containing multiple microchromosomes were determined by analyzing more than 200 spreads from each cell line. (**C**,**D**) Chromosome breaks were evident in ADPKD cells. (**C**) Examples of chromosomes in Giemsa-stained mitotic spreads from HK2 and WT9-12 ADPKD cells. Numerous chromosome breaks can be seen (indicated by arrows). (**D**) The percentages of cells with zero, one, or more than one chromosome break per cell were determined. A total of 100 chromosome spreads were examined for HK2, 100 for WT9-7, and 99 for WT9-12. These findings demonstrate the presence of severe chromosome breakage in ADPKD cells, as indicated by the high frequency of multiple microchromosomes and the presence of chromosome breaks in mitotic spreads.

**Figure 2 ijms-25-02936-f002:**
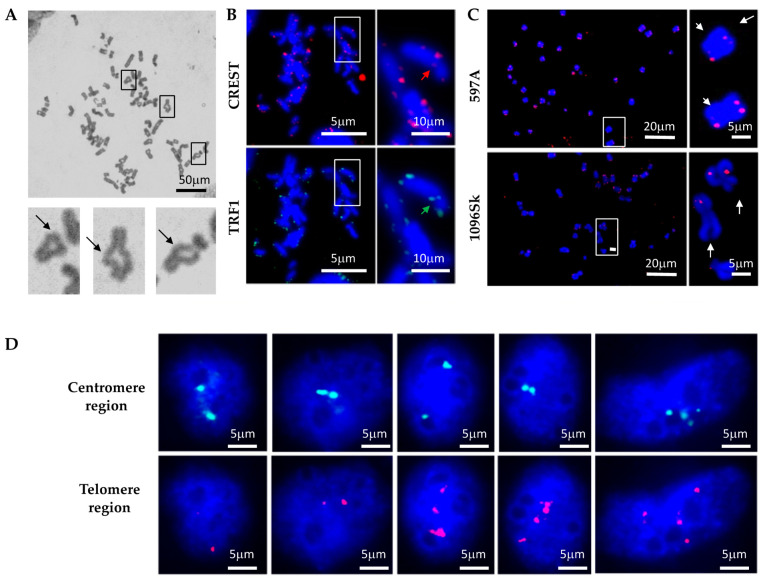
Genomic instability in ADPKD cells. (**A**) Representative examples of fusion chromosomes were observed in primary ADPKD 597A cells. The enlarged insets highlight the fusion of chromosome ends into rings. (**B**) Immunocytochemical staining reveals evidence of chromosome fusion/deletion at centromere and telomere regions. CREST autoimmune sera marked centromeres, while an anti-telomere repeat binding factor 1 (TRF1) antibody labeled telomeres. Representative images are displayed. In each cell type, the lack of CREST antigen in the sister chromatid is indicated by the red arrow, and the green arrow marks the presence of TRF1 outside the telomere region. (**C**) Altered telomeres in ADPKD cells. Chromosome spreads were analyzed using fluorescence in situ hybridization (FISH) with a red fluorescence-labeled telomere probe. Representative images from 597A and 1096Sk cells are presented. The white arrows indicate the loss of one telomere in multiple ADPKD cell chromosomes. (**D**) In situ examination of HK2, 597A, and 1096Sk cells using fluorescent-labeled DNA probes for Smith–Magenis (17P11.2, proximal to centromere, green) and Miller–Diek-r/ILS (17P13.3, proximal to telomere, red) regions of chromosome 17. Following fluorescence in situ hybridization (FISH) staining, nuclei were counterstained with DAPI. Representative examples from 1096Sk cells are shown. Aberrant numbers of green or red dots within individual nuclei are evident.

**Figure 3 ijms-25-02936-f003:**
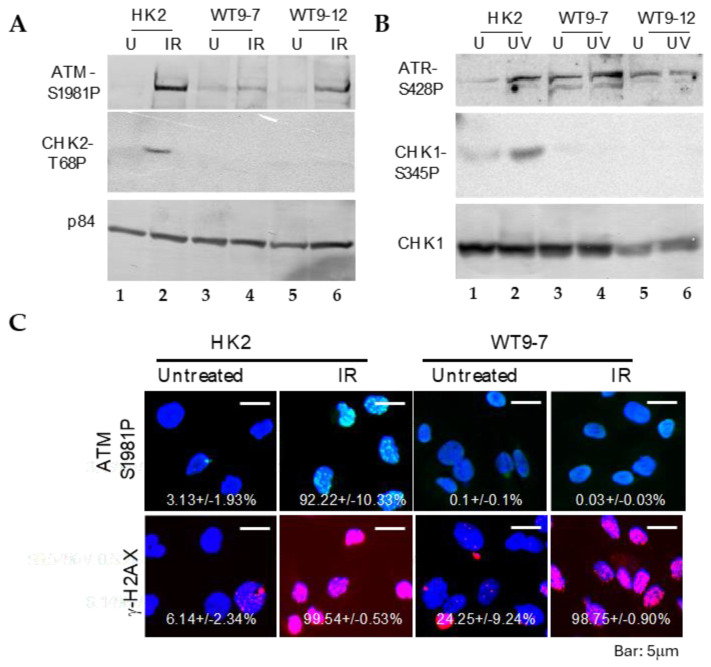
DNA damage-response defect and failure of checkpoint activation in transformed ADPKD cells. (**A**,**B**) Phosphorylation-dependent activation of ATM and ATR, but not CHK1 or CHK2, in response to DNA damage. One hour after exposure to ionizing radiation (IR, A) or ultraviolet radiation (UV, B), ADPKD cells exhibited phosphorylation-mediated activation of ATM at Ser1987 (**A**) and ATR at Ser428 (**B**). However, phosphorylation-dependent activation of CHK1 at Ser345 (**B**) and CHK2 at Tyr68 (**A**) was not observed in the same ADPKD cells. p84 was included as a loading control. (**C**) Failure of IR to induce ATM nuclear foci (IRIF) in ADPKD cells. One hour after IR, cells were fixed and immunostained with anti-phospho-S1987-ATM or anti-γH2AX antibodies. γH2AX nuclear foci, which indicate DNA damage sites [19], were detected in both HK2 and WT9-7 ADPKD cells. In contrast, ATM IRIF were observed in HK2 cells but not in WT9-7 cells. Representative microscopic fields and the mean percentages of cells (± SEM) in each condition containing nuclei with more than five IRIF are shown. Over 200 cells were examined and scored in three separate experiments. The presence of γH2AX nuclear foci in untreated WT9-7 cells is worth noting; it indicates spontaneous DNA damage in these ADPKD cells.

**Figure 4 ijms-25-02936-f004:**
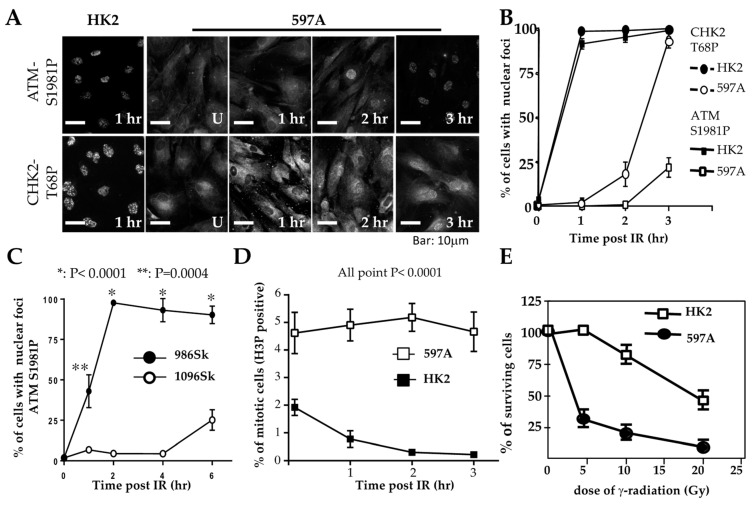
Delayed checkpoint activation in primary ADPKD cells. (**A**) HK2 and 597A cells were left untreated or exposed to γ-radiation (IR, 10 Gy). Cells were then fixed at the indicated time points and immunostained with primary antibodies against phosphorylated ATM-S1981 and phosphorylated CHK2-T68. Representative microscopic fields are shown. (**B**,**C**) Comparative graphs depicting the time courses of activated ATM-S1897-P and activated CHK2-T68-P in nuclear foci after IR in renal cells (HK2 and 597A) over 0–3 h (panel (**B**)) and in skin fibroblasts (986Sk and 1096Sk) over 0-6 h (panel (**C**)). As described in the legend for Figure 3, more than 200 cells of each type were analyzed for nuclei containing > 5 IRIF. The percentages of cells (±SEM) were determined from three independent experiments for each time point. (**D**) M-phase checkpoint analysis. HK2 and 597A cells were treated with IR at time zero, then fixed at hourly intervals and stained with anti-phospho-H3 antibodies. G2/M phase cells were identified as those positive for phospho-H3. The percentages of positively stained cells (±SEM) were scored from more than 400 cells per cell line in duplicate experiments. (**E**) Hypersensitivity of ADPKD cells to IR. HK2 and 597A cells were exposed to the indicated doses of IR. Twenty-four hours after IR, surviving cells were determined by the exclusion of trypan blue. The percentages of surviving cells were compared to control cells that were not irradiated.

**Figure 5 ijms-25-02936-f005:**
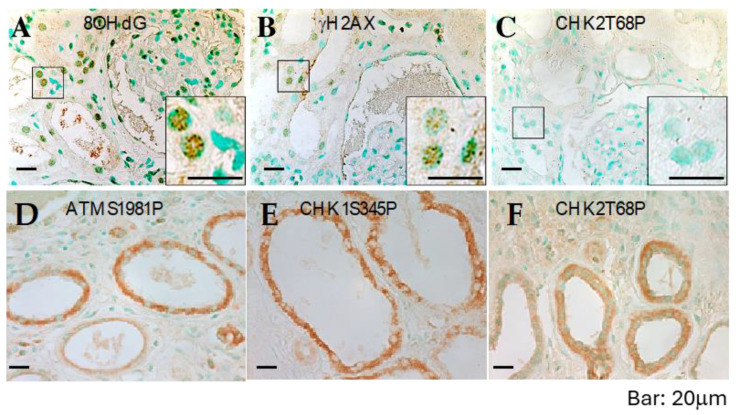
DNA damage and defective response in ADPKD kidneys. Sections from formalin-fixed, paraffin-embedded ADPKD nephrectomy specimens were immunostained with anti-8OHdG, anti-γH2AX, anti-CHK2-T68P, anti-CHK1-S345P, and anti-ATM-S1987P antibodies (all shown), as well as with anti-ATR, anti-Mre11, anti-NFBD1/MDC1, anti-53BP1, and anti-pSQ antibodies. Representative microscopic fields are displayed. Punctate nuclear staining with 8-OHdG antibodies (**A**) and γH2AX antibodies (**B**) can be observed. However, there is minimal or no specific nuclear staining for phosphorylated CHK2-T68P (**C**) or cytosolic staining for phosphorylated ATM-S1987P (**D**), phosphorylated CHK1-S345P (**E**), or phosphorylated CHK2-T68P (**F**). Staining results are summarized in Table 1.

**Figure 6 ijms-25-02936-f006:**
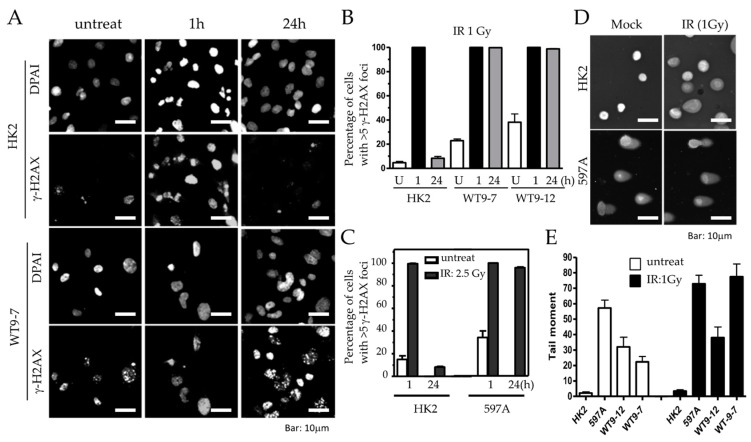
Excessive and persistent DNA damage in ADPKD cells. (**A**) HK2 and ADPKD cells were either left untreated or exposed to low dose of γ-radiation (IR, 1 Gy). After 1 or 24 h, cells were fixed and immunostained with a primary antibody against γH2AX and an Alexa 594 red-tagged secondary antibody. Representative microscopic fields are presented. (**B**,**C**) Histograms illustrating the mean percentages of nuclei (±SEM) with more than five nuclear IRIF (ionizing radiation-induced foci) before and at different time points following irradiation (1 Gy or 2.5 Gy). The histograms were generated from two independent experiments, with more than 200 cells analyzed per time point. (**D**) Representative examples of single cells subjected to alkaline gel electrophoresis and stained with SYBR Green to visualize fast-migrating, damaged DNA in the “comet” tails (Comet Assay™) [26,27]. (**E**) Histograms depicting the quantification of the composite measurement of relative comet tail length, density, and fraction of total DNA in the tail, collectively referred to as “comet tail moment”. The values were normalized to the control (untreated cells).

**Figure 7 ijms-25-02936-f007:**
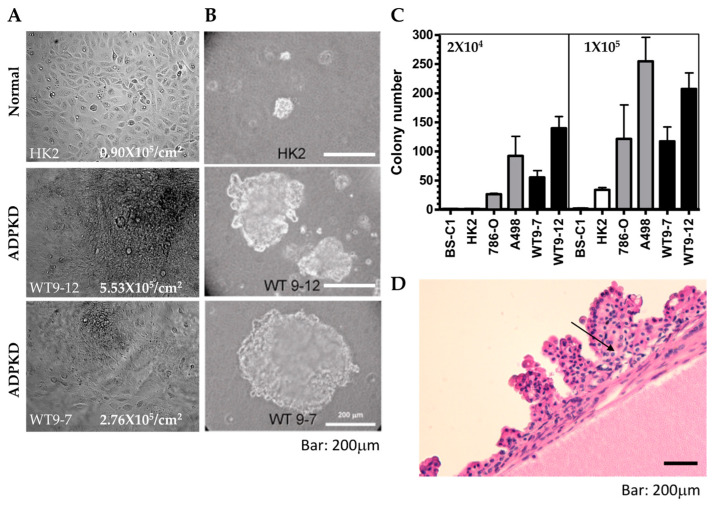
Transformed phenotypes in ADPKD. (**A**) Growth properties of HK2 and ADPKD (WT9-7, WT9-12) cells were assessed. Cells were cultured until confluence and then cultured for two more weeks. While no further proliferation was observed in wild-type HK cells, ADPKD cells grew beyond monolayers, forming piled structures. Representative cell morphology immediately before harvest is shown in the photographs. The saturation density of cells was quantified as the number of cells per cm^2^ on the plates. (**B**,**C**) Anchorage-independent growth of ADPKD cells in soft agar was evaluated. Equal numbers of unclumped cells (high density = 1 × 10^5^/60 mm dish; low density = 2 × 10^5^/60 mm dish) were seeded in duplicate dishes containing 0.367% agar. Total colony numbers were counted after 21 days, and representative colonies were photographed. The scale bar represents 200 μm in (**B**). Histograms depict the neoplastic colony formation of human ADPKD cells, which equal or exceed the colony formation of established renal cell carcinoma lines (786-O and A498). Monkey BS-C1 and human HK2 renal tubular cells served as negative controls. (**D**) Multiple papillary adenomas are commonly found within cysts of ADPKD. Representative photomicrograph is displayed at low magnification (20×).Arrow points to the morphology similar to clear cell renal cell carcinoma.

**Table 1 ijms-25-02936-t001:** Summary of Immunohistochemistry Staining Results in ADPKD Kidney Sections.

Subcelluar Localization	Nuclear with Foci	Nuclear (No Foci)	Cytolytic	Nuclear+ Cytosolic	Undetectable
γH2AX	14/14	0/14	0/14	0/14	0/14
ATMS1981P	1/14	0/14	12/14	1/14	0/14
ATR	0/14	0/14	11/14	3/14	0/14
pSQ	0/14	0/14	2/14	12/14	0/14
CHK1S345P	2/14	0/14	9/14	1/14	2/14
CHK2T68P	2/14	0/14	6/14	2/14	4/14
MDC1/NFBD1	9/14	5/14	0/14	0/14	0/14
53BP1	14/14	0/14	0/14	0/14	0/14
Mre11	3/14	0/14	5/14	1/14	5/14

## Data Availability

Data is contained within the article and Appendix A.

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
