# Peer review of "The Link between Autosomal Dominant Polycystic Kidney Disease and Chromosomal Instability: Exploring the Relationship"

_ijms, 2024, doi:10.3390/ijms25052936_

Round 1
Reviewer 1 Report
Comments and Suggestions for Authors
I read with great interest this study focused on primary renal tubular epithelial cells from human polycystic kidneys that investigates genomic instability and defective DNA damage repair mechanisms involved in ADPKD. Prior studies analyzed the DNA damage repair signaling and found it dysregulated in human ADPKD. I have a few minor issues to be addressed by the authors.
Point 1: Rethinking the abstract would be beneficial, perhaps inserting a few numbers from your results and describing your methods shortly. In its actual form, it is a bit too general.
Point 2 Describe at its first mention within the text HK2 cells.
Author Response
Reviewer 1
I read with great interest this study focused on primary renal tubular epithelial cells from human polycystic kidneys that investigates genomic instability and defective DNA damage repair mechanisms involved in ADPKD. Prior studies analyzed the DNA damage repair signaling and found it dysregulated in human ADPKD. I have a few minor issues to be addressed by the authors.
Point 1: Rethinking the abstract would be beneficial, perhaps inserting a few numbers from your results and describing your methods shortly. In its actual form, it is a bit too general.
Point 2: Describe at its first mention within the text HK2 cells.
Answers:
Thank you for your feedback and the very helpful suggestion about the abstract.
We acknowledge the importance of providing clear and concise descriptions of critical cell lines used in the paper. We did include this information at the first mention of HK2 cells within the text — in the legend for Fig. 1. HK2 cells are described there as "Diploid HPV16 E6/E7 transformed renal tubular epithelial cells (HK2)[12].” The other cell lines used are described in the same legend: “SV40-T antigen transformed ADPKD renal tubular epithelial cells (WT9-7, WT9-12)[13], and untransformed ADPKD skin fibroblast cells (1096Sk).”
We also substantially rewrote the abstract and incorporated more specific details about the most important results. In doing so, we made the abstract more concise and less generic. We hope the reviewer agrees.
Reviewer 2 Report
Comments and Suggestions for Authors
The manuscript prepared by Phang-Lang Chen, Chi-Fen Chen, Hugo Y.-H Lin, Daniel J Riley, Yumay Chen presents a comprehensive study on the association between autosomal dominant polycystic kidney disease (ADPKD) and chromosomal instability, with particular emphasis on genomic instability and defective repair of DNA damage in ADPKD cells. The study covers various aspects such as severe chromosome damage, impaired checkpoint responses and transforming potential of ADPKD cells, highlighting the importance of defective DNA repair in ADPKD pathology.
The study used various techniques, including immunohistochemistry, Comet assays, and examination of cell culture models, to investigate the mechanisms of DNA damage and repair in ADPKD cells. Using both transformed cell lines and primary cells from ADPKD patients provides a solid basis for the findings.
The study provides key information on the DNA damage response and repair pathways in ADPKD, in particular, the role of PC1/PC2 proteins. These findings not only deepen our understanding of the pathogenesis of ADPKD but also suggest new opportunities for therapeutic intervention.
Potential Clinical Implications: Highlighting the link between defective DNA repair mechanisms in ADPKD cells and features commonly found in cancer cells opens up potential therapeutic targets and treatment strategies for ADPKD.
However, despite the good overall assessment of the prepared work, it still has areas requiring improvement:
The first issue is the authors' broader description of the limitations of cell lines. The study used transformed cell lines, which may introduce artifacts that are not representative of primary ADPKD cells. Although using patient-derived primary cells partially addresses this issue, further validation in models with greater physiological relevance would strengthen the conclusions.
Another issue concerns a more detailed analysis of repair paths. The research provided provides us with observations on defective DNA repair in ADPKD cells. However, in my opinion, a more in-depth analysis of specific repair pathways and their interactions may provide a clearer understanding of the underlying mechanisms.
The third aspect is the authors' mention of the so-called “third hit” hypothesis related to environmental factors contributing to cyst formation in ADPKD. Could the authors describe this topic in more depth and expand this aspect to include more detailed interactions of environmental factors with the observed genomic instability? This would shed a broader picture of the pathogenesis of ADPKD.
To sum up
The article makes a significant contribution to the field of ADPKD research by shedding light on the link between genomic instability and the disease. It opens up new research directions and potential therapeutic targets. However, addressing the mentioned limitations could further increase the impact of the results and their translation into clinical practice.
Author Response
Reviewer 2
The manuscript prepared by Phang-Lang Chen, Chi-Fen Chen, Hugo Y.-H Lin, Daniel J Riley, Yumay Chen presents a comprehensive study on the association between autosomal dominant polycystic kidney disease (ADPKD) and chromosomal instability, with particular emphasis on genomic instability and defective repair of DNA damage in ADPKD cells. The study covers various aspects such as severe chromosome damage, impaired checkpoint responses and transforming potential of ADPKD cells, highlighting the importance of defective DNA repair in ADPKD pathology.
The study used various techniques, including immunohistochemistry, Comet assays, and examination of cell culture models, to investigate the mechanisms of DNA damage and repair in ADPKD cells. Using both transformed cell lines and primary cells from ADPKD patients provides a solid basis for the findings.
The study provides key information on the DNA damage response and repair pathways in ADPKD, in particular, the role of PC1/PC2 proteins. These findings not only deepen our understanding of the pathogenesis of ADPKD but also suggest new opportunities for therapeutic intervention.
Potential Clinical Implications: Highlighting the link between defective DNA repair mechanisms in ADPKD cells and features commonly found in cancer cells opens up potential therapeutic targets and treatment strategies for ADPKD.
However, despite the good overall assessment of the prepared work, it still has areas requiring improvement:
The first issue is the authors' broader description of the limitations of cell lines. The study used transformed cell lines, which may introduce artifacts that are not representative of primary ADPKD cells. Although using patient-derived primary cells partially addresses this issue, further validation in models with greater physiological relevance would strengthen the conclusions.
Answer:
We appreciate the reviewer's insightful comment regarding the limitations associated with the use of transformed cell lines in our study. While we acknowledge that transformed cell lines may introduce artifacts not fully representative of primary ADPKD cells, we aimed to mitigate this concern by incorporating patient-derived primary cells in our investigation. The 597A renal tubular epithelial cells used in the key studies were cultured directly from a fresh nephrectomy specimen and used in early passages, as described in the first section of Materials and Methods. Untransformed 1096Sk human skin fibroblasts used in supplemental Fig. S1 served as an additional control for the effects of primary PKD gene mutations without the complicating effects of viral oncogenes on DNA damage and DNA damage repair. We also strove to extend our analyses of cultured cells by demonstrating corroborating abnormalities in the most relevant clinical model, human polycystic kidneys. The results we present are consistent.
We certainly agree that further validation in models with physiological relevance could strengthen the conclusions of our study. Nonetheless, we believe the results we present in ADPKD cells with or without the effects of viral oncogenes, and in human kidneys, support our conclusions.
Another issue concerns a more detailed analysis of repair paths. The research provided provides us with observations on defective DNA repair in ADPKD cells. However, in my opinion, a more in-depth analysis of specific repair pathways and their interactions may provide a clearer understanding of the underlying mechanisms.
Answer:
Again, thank you for the thoughtful comment. We appreciate your suggestion regarding a more detailed analysis of specific repair pathways and their interactions in our study on defective DNA repair in ADPKD cells.
In our defense, we did delve into a detailed discussion in the corresponding section of the manuscript. We emphasized the significance of the observed checkpoint defect, which compromises the timely repair necessary to safeguard the genome. While we acknowledge that a more exhaustive analysis of individual repair pathways and their interplay could provide further insights into the underlying mechanisms, the focus of our study was primarily on highlighting the overarching defects in DNA repair processes within ADPKD cells.
It is our intention to contribute to the scientific understanding of the role of PKD1/PKD2 gene products in the DNA damage response (DDR) pathway. We recognize that our study raises important questions, such as the generalization of PKD1/PKD2 mutations leading to genomic instability, and the specific stage at which polycystins 1 and 2 participate in the DDR pathway. Because we already present a large amount of data in our paper, we decided not to complicate the present study with data that is not as mature. We are still investigating more details about exactly where polycystins fit into DNA damage repair pathways. We will certainly consider your feedback for future research endeavors. With our present manuscript we hope that our findings will inspire other researchers to expand on our work and help us find more answers.
The third aspect is the authors' mention of the so-called “third hit” hypothesis related to environmental factors contributing to cyst formation in ADPKD. Could the authors describe this topic in more depth and expand this aspect to include more detailed interactions of environmental factors with the observed genomic instability? This would shed a broader picture of the pathogenesis of ADPKD.
Answer:
Thank you for bringing up this important topic. We should have made our thinking about the “third hit” clearer. In the revised manuscript we have incorporated a new paragraph to discuss the potential contribution of the "third hit" hypothesis and how understanding the underlying factors could delay the onset of cyst formation in ADPKD.
The new paragraph reads:
"In summary, we have uncovered evidence suggesting that individuals with autosomal dominant polycystic kidney disease (ADPKD) heterozygotes have haploinsufficiency, and with it an impaired capacity to repair DNA damage. This compromised repair state essentially constitutes the second hit, making nullification of the remaining wild-type more likely than usual—and different in each unique cyst within a polycystic kidney. Interestingly, our findings suggest that both the second and third hits required for full manifestation of renal cysts and progressive renal dysfunction may stem from similar threats. Kidney tubular cells, because of their high metabolic rates constant filtration functions, face unique challenges. This "third hit" may manifest in various forms beyond just the need for cell division that has been shown in mouse models with inactivated Pkd1 genes [6]. For instance, compromised repair mechanisms, as observed in conditions like diabetes mellitus, disrupt cellular metabolic homeostasis, leading to dysregulated DNA repair pathways and subsequent genomic instability [25]. Other unique environmental factors, such as the high NaCl-induced exclusion of Mre11 from the nucleus, pose additional hurdles for tubular epithelial cells [42]. Furthermore, hazardous substances like BPA, which are filtered by the kidneys and concentrated in tubular filtrate, pose heightened risks to individuals with compromised repair capacity [43]. Therefore, our observation of defective DNA damage response (DDR) in ADPKD not only sheds light on disease development but also identifies pertinent harmful factors. This insight provides potential avenues for improved care aimed at delaying the progression of ADPKD."
We believe this expanded discussion provides a more comprehensive understanding of the interactions between environmental factors and genomic instability in ADPKD pathogenesis. Thank you again for this suggestion to improve the manuscript. We hope this addition enhances the depth of our analysis sufficiently.
In summary:
The article makes a significant contribution to the field of ADPKD research by shedding light on the link between genomic instability and the disease. It opens up new research directions and potential therapeutic targets. However, addressing the mentioned limitations could further increase the impact of the results and their translation into clinical practice.
Reviewer 3 Report
Comments and Suggestions for Authors
In this report, the authors show that polycystic kidney disease cells behave like primary cancer cells. They lose the ability to activate the DNA damage checkpoint, they have persistent DNA damage and they lose contact inhibition. The authors use a batter of established DNA damage assays to show this. The paper is well written and the experiments support the hypothesis and conclusions.
Author Response
Reviewer 3
In this report, the authors show that polycystic kidney disease cells behave like primary cancer cells. They lose the ability to activate the DNA damage checkpoint, they have persistent DNA damage and they lose contact inhibition. The authors use a battery of established DNA damage assays to show this. The paper is well written and the experiments support the hypothesis and conclusions.
Answer/Comment:
Thank you for recognizing our efforts in highlighting the significance of defective DNA damage repair in autosomal dominant polycystic kidney disease (ADPKD). We are glad to learn that our paper effectively demonstrates the parallels between ADPKD cells and primary cancer cells, shedding light on their shared characteristics such as compromised DNA damage checkpoint activation, persistent DNA damage, and loss of contact inhibition. Your feedback is greatly appreciated, and we are pleased that our experiments support the hypothesis and conclusions outlined in the manuscript.
Round 2
Reviewer 2 Report
Comments and Suggestions for Authors
Dear Authors,
Thank you for your response and allaying my concerns about the manuscript. In my opinion, the revised work meets the criteria of a scientific work and its clinical significance is well described.
In my opinion, the current form of the article meets the standards for the IJMS magazine. I look forward to further results of your work and wish you good luck.